# $NO_3^-$ anions can act as Lewis acid in the solid state

Antonio Bauzá[1], Antonio Frontera[1] & Tiddo J. Mooibroek[2,3]

Identifying electron donating and accepting moieties is crucial to understanding molecular aggregation, which is of pivotal significance to biology. Anions such as $NO_3^-$ are typical electron donors. However, computations predict that the charge distribution of $NO_3^-$ is anisotropic and minimal on nitrogen. Here we show that when the nitrate's charge is sufficiently dampened by resonating over a larger area, a Lewis acidic site emerges on nitrogen that can interact favourably with electron rich partners. Surveys of the Cambridge Structural Database and Protein Data Bank reveal geometric preferences of some oxygen and sulfur containing entities around a nitrate anion that are consistent with this 'π-hole bonding' geometry. Computations reveal donor–acceptor orbital interactions that confirm the counterintuitive Lewis π–acidity of nitrate.

[1] Department of Chemistry, Universitat de les Illes Balears, Carretera de Valldemossa km 7.5, 07122 Palma, Baleares, Spain. [2] Faculteit der Natuurwetenschappen, Wiskunde en Informatica, van 't Hoff Institute for Molecular Sciences, Universiteit van Amsterdam, Science Park 904, 1098 XH Amsterdam, The Netherlands. [3] School of Chemistry, Faculty of Science, University of Bristol, Cantock's Close, Bristol BS8 1TS, UK. Correspondence and requests for materials should be addressed to A.F. (email: toni.frontera@uib.es) or to T.J.M. (email: t.j.mooibroek@uva.nl).

Molecular recognition phenomena are of pivotal significance in biology and define the field of supramolecular chemistry[1–4]. Well-known intermolecular forces include hydrogen and halogen bonding. Both have recently been contextualized as instances of 'σ-hole bonding'[5–7]. A σ-hole can be seen as a region of electropositive potential on a molecule that is roughly located on the unpopulated σ* antibonding orbital of a covalent bond. Typical examples of such σ-holes can be found along the O–H/C–Br vectors in phenol or bromobenzene[8]. In analogy, a π-hole can be seen as a region of electropositive potential on a molecule that is roughly located on an unpopulated π* antibonding orbital of a π bond, for example, on carbonyls or π-acidic aromatics like hexafluorobenzene[8–11].

It is known that π-holes in nitro-compounds such as nitrobenzene (Fig. 1a) can be directional in the solid state[8,12]. The magnitude of such π-holes can be enhanced when the negative charge is diluted over a larger area, for example, if the O-atoms interact with water or NaCl (Fig. 1b,c). We wondered to what extend this rationale applies to nitrate anions and if perhaps this anion might function as a π-hole to enable so-called (pseudo)anti-electrostatic interactions[13,14]. This seems counterintuitive, yet the charge distribution in naked $NO_3^-$ is anisotropic and reminiscent of a π-hole (Fig. 1d)[15]. Dampening the charge with water further exposes the π-hole (Fig. 1e), and the potential even becomes positive at $+25$ kcal mol$^{-1}$ in charge-neutral [$LiNO_3 \cdot 2H_2O$] (Fig. 1f). We found that such an exposed π-hole on $NO_3^-$ can form complexes with electron rich partners with calculated energies of up to $-31.6$ kcal mol$^{-1}$. Statistical evaluations of the Cambridge Structural Database (CSD) and the Protein Data Bank (PDB) indeed reveal some geometric preferences consistent with this 'π-hole bonding' geometry. Several examples lifted from these databases are highlighted, where the Lewis acidity of $NO_3^-$ seems evident.

## Results

**Computational models**. To further evaluate a possible π-hole on $NO_3^-$ computationally, we designed trisurea based receptors that fully surround a nitrate anion with hydrogen bond donors (Fig. 2a, related [2+2] macrocycles are known)[16,17]. This might mimic nitrate anions in crystal structures, which are typically enclosed by several (charge assisted) hydrogen bonds and/or charge compensated by coordination to a metal ion. In the anionic complexes **2** (Fig. 2b), the π-hole region represents a relative electron depletion and is positive at $+25$ kcal mol$^{-1}$ in the anionic species **2c**.

Next, we computed some complexes of **2** with electron rich partners and found that complexes [**2c**···NCCH₃]$^-$ and [**2c**···Cl]$^{2-}$ gave interaction energies of $-7.7$ and $-31.6$ kcal mol$^{-1}$, respectively (see Supplementary Note 1 and Supplementary Tables 1 and 3, for details). These values are in the

range of weak to strong hydrogen bonding[18]. An 'atoms-in-molecules' (AIM) analysis[19] revealed a single-bond critical point between the nitrate N-atom and N/Cl of the interacting partner (see Supplementary Figs 1 and 2).

**Database analyses**. Encouraged by these computational predictions we wondered if there might be any experimental evidence for π-hole bonding with $NO_3^-$ within the crystal structures deposited in the CSD[20] and the Brookhaven PDB[21]. More in particular we were interested in ascertaining any possible directional behaviour[22] of this interaction. We limited our inquiries to uncoordinated nitrate anions to simplify our analysis. Also, if uncoordinated nitrate can act as Lewis acid, it is likely that coordinated nitrate will do so as well (and likely even more so). Initial data sets were retrieved by limiting the $^-O_3N\cdots$El.R. distance to 5 Å (El.R. = 'electron rich atom'). The data are thus confined within a sphere with 5 Å radius but will—due to symmetry—be represented as contained within a 5 Å high and 10 Å wide hemisphere. The interacting entities considered here are $H_2O$, O=X (X = any atom for the CSD data and only C for the PDB data) and S (CSD)/S–C (PDB). Further details of the methods employed can be found in the 'Methods section'.

Shown in Fig. 3 are the distributions in three dimensional space of O-atoms in $H_2O$ (left of panel), O=X/C (middle of panel) and of S-atoms (right of panel) around a nitrate anion as found within the CSD (top of panel) and the PDB (bottom of panel). These distributions are remarkably similar in both databases. Water molecules seem to cluster near the O-atoms and are relatively in-plane with $NO_3^-$, suggesting that hydrogen bonding dominates. On the other hand sp²–hybridized O-atoms and S-atoms tend to cluster perpendicular to the $NO_3^-$ plane, above/below the nitrate's N-atom. This clustering is consistent with a π-hole bonding geometry. Four dimensional (4D) density plots (Supplementary Fig. 3) further confirm these trends and directionality plots clearly indicate that $NO_3^-$ π-hole bonding is about as directional as NH···π$^{aryl}$ hydrogen bonding (Supplementary Fig. 4)[8,22–25]. Also, significant amounts of data directly above/below N (13–21%) consist of overlapping van der Waals shells (Supplementary Fig. 5).

**Concrete examples**. Finally, we selected several examples of crystal structures displaying short $^-O_3N\cdots$El.R. distances and geometries consistent with a π-hole interaction (El.R. = 'electron rich atom'). Shown in Fig. 4 (top) are charge-neutral selections of the small salts EVIKEA[26], ORUHOZ[27] and BIDHAX[28] lifted from the CSD. In all these instances the $^-O_3N\cdots$El.R. distances are within the sum of the van der Waals radii of the elements involved and the nitrate anion is concurrently entrenched in a hydrogen bonding pocket (not shown).

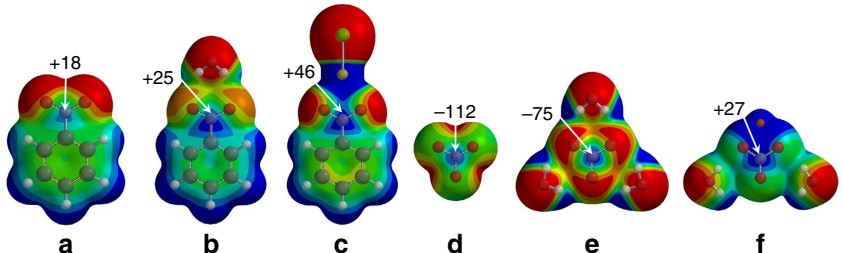

**Figure 1 | Some molecular electrostatic potential maps (MEPs) of nitrobenzene and nitrate.** Nitrobenzene (**a**) interacting with water (**b**) and NaCl (**c**) and $NO_3^-$ (**d**) interacting with three water molecules (**e**) or one Li$^+$ and two water molecules (**f**). Geometries were optimized with DFT/BLYP/6-31G* and MEPs and energetic values (in kcal mol$^{-1}$) generated at the MP2/6-311+G** level of theory. The colour codes of the MEPs represent more negative (red) to more positive (blue) potentials in between: $+36$ and $+24$ (**a,b**); 0 and $+48$ (**c**); $-155$ and $-122$ (**d**); $-102$ and $-75$ (**e**); $-48$ and $+27$ (**f**).

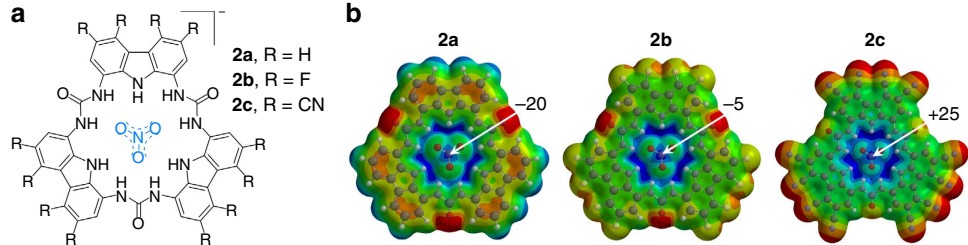

**Figure 2 | Nitrate complexes with trisurea macrocycles.** (**a**) Schematic drawing of **2**. (**b**) MEPs of **2a**, **2b** and **2c**. Geometries were optimized with DFT/BLYP/6-31G* and MEPs and energetic values (in kcal mol⁻¹) generated at the B3LYP/6-31G* level of theory. The colour codes of the MEPs represent more negative (red) to more positive (blue) potentials in between: − 80 and − 18 (**2a**); − 72 and − 1 (**2b**); − 80 and + 35 (**2c**).

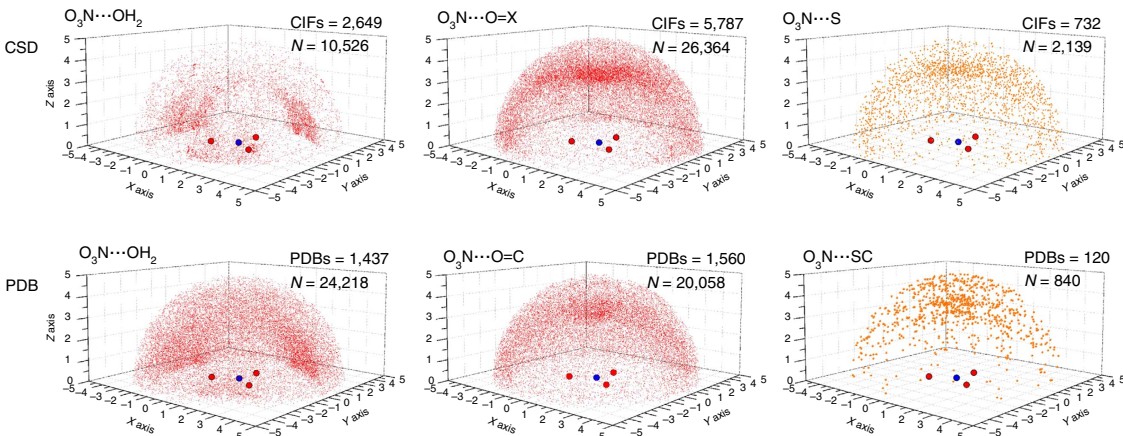

**Figure 3 | Database analyses.** Distribution of the O- or S-atoms (belonging to water (left of panel), OX/OC (middle of panel) or S/SC (right of panel) entities) around an uncoordinated nitrate anion as found within the CSD (top of panel) and PDB (bottom of panel) and contained by the parameter $^{nitrate}N \cdots O/S \leq 5$ Å. X = any atom.

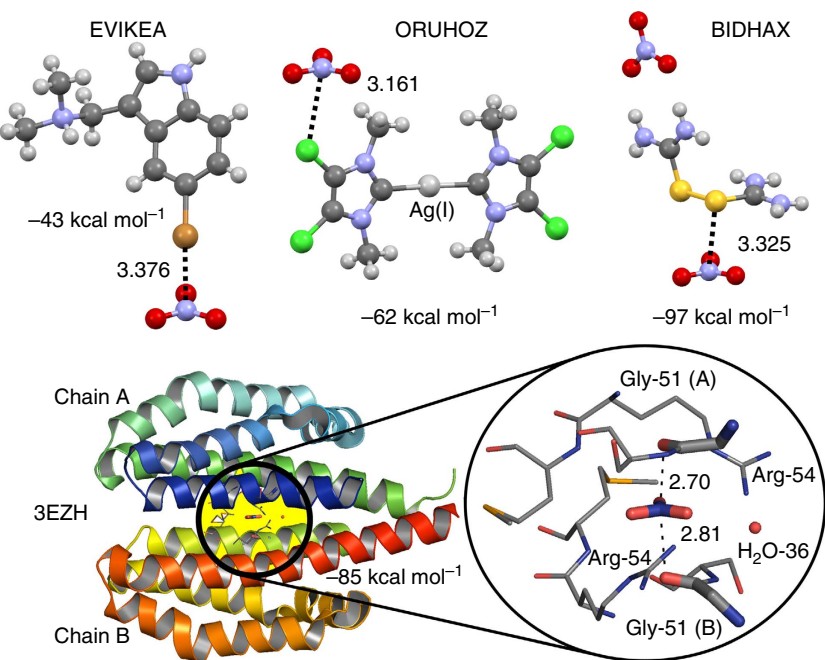

**Figure 4 | Examples of crystal structures exhibiting nitrate π-hole interactions.** Top of panel: three structures found in the CSD. Bottom of panel: example lifted from the PDB with a zoom-in of the nitrate ligand's binding pocket (residues ≤ 4 Å displayed). All these selected fragments were computed at the BP86-D3/def2TZVP level of theory leading to the indicated energies (dominated by charge compensation). Colour code: carbon = grey, hydrogen = white, nitrogen = blue, oxygen = red, sulfur = yellow, chloride = green, bromide = brown and silver = light grey. See Supplementary Table 3 for Cartesian coordinates of selected (and computed) fragments.

Protein structure 3EZH (ref. 29; Fig. 4, bottom) consists of two isostructural chains (A and B) that are stuck together surrounding a central nitrate ligand. The $NO_3^-$ anion is H-bonded to two arginine residues (Arg-54) with $N\cdots N$ distances of about 3 Å. In addition, the carbonyl O-atoms of two glycine residues (Gly-51) appear to interact with the nitrate ligand's π-hole. Indeed, the interatomic $N\cdots O$ distances (2.70 and 2.81 Å) are well within the van der Waals benchmark for $N+O$ (3.07 Å).

The AIM analyses of these four examples revealed a clear bond critical point between the nitrate's N-atom and the interacting electron donor for both EVIKEA and 3EZH (Supplementary Fig. 6). A natural bond orbital analysis (with a focus on second order perturbation)[30] for each example revealed electron donation from a lone pair of electrons (LP) into an unoccupied orbital of the nitrate anion (for example, 0.35 kcal mol$^{-1}$ for the $LP(C=O)\rightarrow\pi^*(NO_3^-)$ in 3EZH, see Supplementary Note 2 for details). This confirms that the $NO_3^-$ anion can act as a Lewis acid in these examples.

## Discussion

The above computations, database analyses and examples clearly point out that a genuine π-hole might persist on a nitrate anion and that $NO_3^-$ may thus act as a Lewis acid to form formally (pseudo)anti-electrostatic interactions in the solid state. One naturally wonders what other anions might be capable of displaying such behaviour (using other atoms than hydrogen). It occurred to us that nitrate actually has a rather unique set of properties that set it apart from other common anions in this respect: $NO_3^-$ is fairly polarized and further polarizable, not so charge-dense and nitrate is flat, rendering the π-hole sterically accessible (see Supplementary Note 3 for a discussion of possible candidates). As nitrate anions are very common in chemistry and biology, we anticipate that our finding may serve as a (retrospective) guide to interpret chemical data where nitrate anions are involved; for example, orthonitrate formation[31,32], cases where $NO_3^-$ anions may be a structural determinant (as in 3EZH), or in transport phenomena involving this ubiquitous anion[33,34].

## Methods

**Computations.** The energies of all complexes included in this study were computed at the BP86-D3/def2-TZVP level of theory. The calculations have been performed by using the programme TURBOMOLE version 7.0 (ref. 35). For the calculations we have used the BP86 functional with the latest available correction for dispersion (D3)[36]. The optimization of the molecular geometries has been performed imposing the $C_{3v}$ symmetry point group. The Bader's 'Atoms in molecules' theory has been used to study the interactions discussed herein by means of the AIMall calculation package[37].

**Queries used to retrieve data from the CSD and PDB.** The CSD (version 5.37 (November 2015 including two updates) was inspected using ConQuest (version 1.18) on the 3rd of April 2016. The PDB was inspected with the online Query Sketcher of Relibase version 3.2.1 on the 2nd of March 2016. For the CSD search, a subset of data was first created containing uncoordinated nitrate anions (9,439 crystallographic information files). All searches of the CSD were limited to high quality structure ($R \leq 0.1$) and powder structures and structures containing errors were omitted. The N–O bonds were set to 'any type'. All covalent bond distances and selected triatomic angles were collected to reconstruct the average models (one for the PDB data and one for the CSD data) used for accessing directionality (see below). The interatomic distance between the interacting atom (O in O=C or OH$_2$; S in S, SC or SCC; or F, Cl, Br, I, At, O, S, Se, Te, N, P or As in El.R. ('electron rich') and the nitrate's N-atom (e, highlighted in red in Fig. 5) was set as $\leq 5$ Å so that the data was confined within a 10 Å diameter sphere centred on N. In the PDB study the NO$_3$ central unit was marked as a ligand and the interacting atom(s) were marked as part of a protein, or in the case of water the interacting O was specified as water.

**Deriving XYZ coordinates and r.** The interatomic distances between the interacting atom and O1 and O3, as well as the O1–O3 distance were also collected (set to $\leq 8$ Å in the PDB search and left unspecified for the CSD search). The triangle formed by O1–N–O3 was chosen as the base, and the interacting atom as

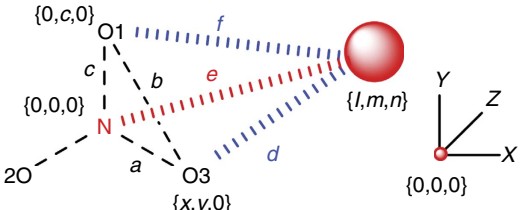

**Figure 5 | Query used for PDB and CSD search.** Relevant interatomic distances (e.g. a–f) and triatomic angles (e.g. O1-N-O3) were retrieved from the databases in order to construct a model of the central $NO_3^-$ anion and to obtain the Cartesian coordinates (l,m,n) of the interacting atom (large red sphere) as is detailed in the methods section. The XYZ axis on the right is meant as a guide to the eye, centered on {0,0,0} and with the Y axis in line with N-O1.

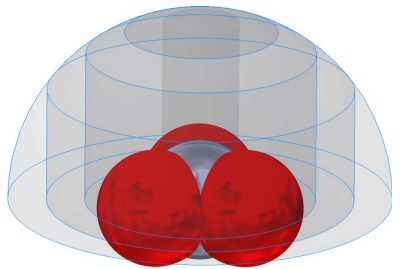

**Figure 6 | Nitrate model and the volumes used to access directionality.** The central nitrate model was constructed with Autodesk Inventor Professional 2016 using the average distances/angles as observed in a database and with literature van der Waals radii. The gray bodies illustrate the cylindrically trimmed hemispheres at $\mathbf{r} = 1, 2, 3, 4,$ and 5 Å used to assess directionality.

the tip of a tetrahedron (see Fig. 5) so that Cartesian Coordinates {X, Y, Z} of all the atoms could be derived as follows: the N-atom was taken as the centre {0, 0, 0}, O1 as {0, c, 0}, O3 as {x, y, 0} and the interacting atom at {l, m, n}. Distances a–f were measured, from which y, x, m, l and n can be derived using equations (1–5), respectively.

$$y = \frac{a^2 + c^2 - b^2}{2c} \tag{1}$$

$$x = \sqrt{a^2 - y^2} \tag{2}$$

$$m = \frac{c^2 + e^2 - d^2}{2c} \quad (=X\text{-value}) \tag{3}$$

$$l = \frac{a^2 + e^2 - f^2 - 2my}{2x} \quad (=Y\text{-value}) \tag{4}$$

$$n = \sqrt{e^2 - m^2 - l^2} \quad (=Z\text{-value}) \tag{5}$$

Thus, the distance between the interacting atom and the plane defined by O1–N–O3 is $n$, that is, the Z-value. With this and the $N\cdots$ interacting atom distance ($e$) the parallel displacement parameter ($\mathbf{r}$) could be derived according to equation (6):

$$\mathbf{r} = \sqrt{e^2 - n^2} \tag{6}$$

With this procedure the sign of $n$ (that is, the Z-axis) is always positive, meaning that data in one half of the sphere were reflected to the other half of the sphere to obtain the data within a 5 Å high and 10 Å wide hemisphere. To obtain all {X, Y, Z} coordinates of the average models, it was assumed that O2 was coplanar with O1–N–O3. The averages of relevant distances and angles were then used together with the rules of sine and cosine to obtain the {X, Y, Z} coordinates. The relative standard deviations of the parameters used were typically below 5%. A numerical overview of the data retrieved is shown in Supplementary Table 2.

**Rendering 4D plots and analysis of directionality.** 4D density plots were generated by first binning the data (using a custom build Excel spreadsheet, available on request) in 405 volumes $\{X\ [9 \times {}^{10}/_9\ \text{Å}],\ Y\ [9 \times {}^{10}/_9\ \text{Å}],\ Z\ [5 \times {}^{5}/_5\ \text{Å}]\}$. The percentage of the total that each volume contains was computed by dividing the number of data in a certain volume by the total amount of data. This density information was projected onto the centre of each volume using Origin Pro 8. The size and colour of the spheres in the resulting plots are a visual representation of the density of data, whereby red and larger is denser, empty and small is less dense.

The average $\{X, Y, Z\}$ coordinates of the atoms of nitrates found within the CSD or the PDB, together with the standard van der Waals radii for N (1.55 Å) and O (1.52 Å) were used to generate a model as a single body 'part' file (.ipt) using Autodesk Inventor Professional 2016 (by using mm instead of Å). Similarly, a hemisphere was created with a radius of 5 mm. Derived from this hemisphere were bodies where the volume above the base with basal radius (representative for **r**) was trimmed, that is, 'cylindrically trimmed hemispheres'.

The $NO_3$ model, the hemisphere and the cylindrically trimmed hemispheres were collected in an assembly file (.iam), properly alighted, as is illustrated in Fig. 6 with (cylindrically trimmed) hemispheres of 1, 2, 3, 4 and 5 mm basal radius.

Using the 'Analyse Interference' option in Autodesk Inventor Professional 2016 the interfering volumes between the model and the cylindrically trimmed hemispheres could be determines. The volume difference between two such interfering volumes of incremental **r**-values, say $\mathbf{r}_a$ and $\mathbf{r}_b$, thus represent the volume that the model occupies in between two values of **r**, that is, $V_{model}$. Similarly, the interfering volume between two cylindrically trimmed hemispheres could be derived as a function of **r**, from which the volume in between two **r**-values as found within the hemisphere could be derived, that is, $V_{no\ model}$. The actual free volume in between two **r**-values that a 'host' can occupy, that is, $V_{free}^r$, is thus given $V_{no\ model} - V_{model}$. The total freely accessible volume, $V_{free}^{total}$, is naturally given the volume of the hemisphere minus the volume of the model in the hemisphere. The random (or volume) distribution as a function of **r**, that is, $D_{chance}^r$, is thus given by:

$$D_{chance}^r = \frac{V_{free}^r}{V_{free}^{total}} \qquad (7)$$

The actual distribution of the data, $D_{data}^r$, is naturally given by:

$$D_{data}^r = \frac{N^r}{N^{total}} \qquad (8)$$

Thus, the chance corrected distribution of data, $P(\mathbf{r})$ is given by:

$$P(\mathbf{r}) = \frac{D_{data}^r}{D_{chance}^r} \qquad (9)$$

For an accidental distribution, $P$ should be unity across all **r**-values; a $P$ value greater than unity is thus evidence of positive clustering (suggesting a favourable interaction), while $P$ values smaller than unity reflect a depletion of data (suggesting an unfavourable interaction).

**Data availability.** All the data that support the findings of this work are available from the corresponding authors upon reasonable request.

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

## Acknowledgements

A.F. and A.B. thank the MINECO of Spain (projects CTQ2014-57393-C2-1-P and CONSOLIDER INGENIO 2010 CSD2010-00065, FEDER funds) for funding. We also thank the 'Centre de Tecnologies de la Informació' (CTI) at the UIB for computational facilities. T.J.M. partially conducted the work with funds from the research programme 'VIDI' with project number 723.015.006, which is financed by the Netherlands Organisation for Scientific Research (NWO).

## Author contributions

Most of the computational studies were conducted by A.B. and A.F., some by T.J.M. The database analyses were conducted by T.J.M., A.F. and T.J.M. wrote the article and directed the study.

## Additional information

**Competing financial interests:** The authors declare no competing financial interests.

**Publisher's note**: 

