## [Peer Review File · Nature Communications]

Reviewers' Comments:

Reviewer #1 (Remarks to the Author)

I found this to be a compelling analysis of the electronic distribution in nitrate combining modeling studies with an analysis of the geometry of the interactions of other entities with nitrate on the Cambridge Crystallographic Database and the Brookhaven PDB. The results are quite clear that water clusters around the oxygenated through hydrogen bonding O= atoms and S cluster above the pi-hole. This is an important finding as it gives a new way of complexing nitrate - a notorious difficult ion to bind due to its weak basicity.

Reviewer #2 (Remarks to the Author)

The communication is a significant work using pi-hole bonding model describe some behaviours of NO₃⁻ in solid state, especially in coordination compounds or some supramolecular entities. I have searched some published papers via CSD, for example, *Inorg. Chem.* 2014, 53, 11749–11756; *J. Coord. Chem.* 2013, 66:21, 3782-3790; *Inorg. Chem.* 2013, 52, 13791–13802, and so on, and in these papers, -3ON---O interaction indeed occurs and I believe the interaction should contribute to the stability of coordination compounds although the authors did not concern on the interaction besides -3ON---M coordination. I think that publication of the submitted communication could pay attention to role of nitrate in coordination chemistry.

However, in both originality of concept and importance of NO₃⁻ in biological and chemical aspects the communication is not appropriated to be published in the Nature communication.

Two questions are that (1) the fact described in MS "computations predict that the charge distribution of NO₃⁻ is anisotropic and minimal on N" has been appeared in text book, not now, for example, C. E. Housecroft, A. G. Sharpe. *Inorganic Chemistry* (2nd ed). Prentice Hall. 2005, page 106-107, page 120.

(2) The title is not appropriated and is easily misleading. Because in free state, NO₃⁻ anion cannot act as Lewis acid, but when the nitrate's charge is sufficiently dampened by resonating over a larger area, a Lewis acidic site emerges on N that can interact favorably with electron rich partners, as described in the MS.

The *J Phys Chem B* or *PCCP* is more appropriated to publish the work.

Reviewer #3 (Remarks to the Author)

In general this communication concerns the Lewis acid properties of the NO₃⁻ anion.

The first part of this study refers to computational results;

it is shown that NO₃⁻ not involved in any interactions is characterized by the negative EP at the nitrogen center; next if oxygen atoms interact with the electron withdrawing species the EP at N-center is modified, and it may be positive thus may play a role of the Lewis acid center;

it is a very important result and that inspired the authors of this communication to perform searches through CSD and PDB bases;

it was done and the possible Lewis acid properties were justified since there are numerous crystal structures where they are revealed.

The manuscript is clear, and the findings are important and interesting for the broader community of chemists.

The calculations were performed at the sufficient level since the systems analyzed are sometimes

large (especially those presented in Fig. 2).

This is why I recommend the publication of this manuscript almost as it stands;
there is only slight suggestion from my side;
first paragraph of the communication;

“A σ -hole can be seen as a region of electropositive potential on a molecule that is roughly located on the unpopulated σ^* antibonding orbital of a covalent bond, e.g. along the O–H/C–Br vectors in phenol or bromobenzene.”

The sigma and pi hole concept is not commonly known for the researchers' community, this paragraph may suggest that these concepts are restricted to the examples presented, hence, in my opinion it should be slightly modified.

Reviewer #1 (Remarks to the Author):

I found this to be a compelling analysis of the electronic distribution in nitrate combining modeling studies with an analysis of the geometry of the interactions of other entities with nitrate on the Cambridge Crystallographic Database and the Brookhaven PDB. The results are quite clear that water clusters around the oxygenated through hydrogen bonding O= atoms and S cluster above the pi-hole. This is an important finding as it gives a new way of complexing nitrate - a notorious difficult ion to bind due to its weak basicity.

Response: We are very grateful to this referee for his/her assessment and his/her recognition of the significance of our finding.

Reviewer #2 (Remarks to the Author):

The communication is a significant work using pi-hole bonding model describe some behaviours of NO₃⁻ in solid state, especially in coordination compounds or some supramolecular entities. I have searched some published papers via CSD, for example, Inorg. Chem. 2014, 53, 11749–11756; J. Coord. Chem. 2013, 66:21, 3782-3790; Inorg. Chem. 2013, 52, 13791–13802, and so on, and in these papers, -3ON---O interaction indeed occurs and I believe the interaction should contribute to the stability of coordination compounds although the authors did not concern on the interaction besides -3ON---M coordination. I think that publication of the submitted communication could pay attention to role of nitrate in coordination chemistry.

Response: The referee comments about coordinated nitrate anions, and we are grateful that (s)he raised this issue. In the early stages of our database research we tried to focus on coordinated nitrate and indeed found many examples of apparent π -hole interactions with M–NO₃⁻ fragments. However, this analysis is complicated by the fact that there are many different metals in various oxidation states and by the different possible coordination modes. Nitrate can be coordinated to 1, 2, 3 or even more metals (μ -coordination) and it can act as a (pseudo) chelate. Moreover, we thought that if we could demonstrate Lewis acidic behaviour of uncoordinated nitrate anions, it would follow that this rationale will also (and likely more so) apply to coordinated nitrate, where the charge is directly compensated. We have now added a comment (highlighted in yellow on page 2/3) to make this point more explicit.

However, in both originality of concept and importance of NO₃⁻ in biological and chemical aspects the communication is not appropriated to be published in the Nature communication.

Two questions are that (1) the fact described in MS "computations predict that the charge distribution of NO₃⁻ – is anisotropic and minimal on N " has been appeared in text book, not now, for example, C. E. Housecroft, A. G. Sharpe. Inorganic Chemistry (2nd ed). Prentice Hall. 2005, page 106-107, page 120.

Response: We did not (indent to) claim that the anisotropic charge distribution of the nitrate anion is our finding, as this is to be expected if one simply draws resonance structures of NO₃⁻. The most extreme resonance structure even 'predicts' a charge of 2+ on N. It must be stressed however, that while the charge *distribution* is obviously anisotropic, the *absolute electronic potentials* are negative on all atoms in naked nitrate. Thus, finding that an NO₃⁻ anion may actually act as a Lewis acid in the solid state (as all three reviewers acknowledge) is counterintuitive and unexpected. We ourselves were initially surprised by the clarity of our database analyses. To avoid any possible misinterpretation of this issue, we have included the reference suggested by this referee (N^o 15 in the revised manuscript, also highlighted in yellow).

(2) The title is not appropriated and is easily misleading. Because in free state, NO₃⁻ anion cannot act as Lewis acid, but when the nitrate's charge is sufficiently dampened by resonating over a larger area, a Lewis acidic site emerges on N that can interact favorably with electron rich partners, as described in the MS.

Response: We are a bit puzzled that this referee thinks that the title may suggest that 'free' NO₃⁻ can act as Lewis acid. 'Free' NO₃⁻ cannot exist *in the solid state* –or in solution– due to charge compensation and unavoidable interactions with neighbouring atoms/molecules. Actually, a truly 'free' NO₃⁻ may only exist in the vacuum of mass spectrometers. It is also very clearly stated both in the abstract and throughout the paper that the charge of NO₃⁻ anions must be *sufficiently* diluted over a larger area in order for nitrate to act as Lewis acid. We could make this explicit in the title, but we feel that the title then becomes needlessly lengthy. Note also that the other referees apparently found the title appropriate.

The J Phys Chem B or PCCP is more appropriated to publish the work.

Reviewer #3 (Remarks to the Author):

In general this communication concerns the Lewis acid properties of the NO₃⁻ anion. The first part of this study refers to computational results; it is shown that NO₃⁻ not involved in any interactions is characterized by the negative EP at the nitrogen center; next if oxygen atoms interact with the electron withdrawing species the EP at N-center is modified, and it may be positive thus may play a role of the Lewis acid center; it is a very important result and that inspired the authors of this communication to perform searches through CSD and PDB bases; it was done and the possible Lewis acid properties were justified since there are numerous crystal structures where they are revealed.

The manuscript is clear, and the findings are important and interesting for the broader community of chemists. The calculations were performed at the sufficient level since the systems analyzed are sometimes large (especially those presented in Fig. 2).

This is why I recommend the publication of this manuscript almost as it stands; there is only slight suggestion from my side; first paragraph of the communication; "A σ -hole can be seen as a region of electropositive potential on a molecule that is roughly located on the unpopulated σ^* antibonding orbital of a covalent bond, e.g. along the O–H/C–Br vectors in phenol or bromobenzene." The sigma and pi hole concept is not commonly known for the researchers' community, this paragraph may suggest that these concepts are restricted to the examples presented, hence, in my opinion it should be slightly modified.

Response: We are first and foremost very grateful to this referee for his/her assessment and his/her recognition of the significance of our finding. Secondly his/her suggestion is much appreciated and we have altered the sentence in question accordingly (highlighted in yellow on page 1).

Reviewers' Comments:

Reviewer #2 (Remarks to the Author)

I have checked carefully the responses to my comments and questions as well as the revised MS, and they are satisfactory. Also, I read the comments from other two referees.

Here, I respect editor's decision: that is, the revised MS can be accepted to be published in Nature Communications.